# Stage-Dependent Levels of Brain-Derived Neurotrophic Factor and Matrix Metalloproteinase 9 in the Prognosis of Colorectal Cancer

**DOI:** 10.3390/biomedicines11071839

**Published:** 2023-06-26

**Authors:** Ivana Večurkovská, Jana Mašlanková, Vladimíra Tomečková, Jana Kaťuchová, Terézia Kisková, Lucia Fröhlichová, Mária Mareková, Marek Stupák

**Affiliations:** 1Department of Medical and Clinical Biochemistry, Faculty of Medicine, Pavol Jozef Šafarik University in Košice, Trieda SNP 1, 040 11 Košice, Slovakia; 21st Department of Surgery, Faculty of Medicine, Pavol Jozef Šafarik University in Košice, Trieda SNP 1, 040 11 Košice, Slovakia; 3Department of Animal Physiology, Institute of Biology and Ecology, Faculty of Science, Pavol Jozef Šafarik University in Košice, Trieda SNP 1, 040 11 Košice, Slovakia; terezia.kiskova@upjs.sk; 4Department of Pathology, Louis Pasteur University Hospital, Rastislavova 43, 041 90 Košice, Slovakia; lucia.frohlich@gmail.com

**Keywords:** colorectal cancer, biomarkers, brain-derived neurotrophic factor (BDNF), matrix metalloproteinases (MMPs)

## Abstract

**Simple Summary:**

The ever-increasing number of CRC cases and the associated number of deaths from this type of cancer require the development of sensitive, accurate, and non-invasive biomarkers for the early detection of CRC. At the same time, it is necessary to characterize these biomarkers in relation to the individual stages of CRC. We achieved interesting results by creating a BDNF/MMP9 ratio. The tissue BDNF/MMP-9 ratio (evaluated immunohistochemically) decreased significantly with the progression of the disease in living patients. However, in deceased individuals, the ratio showed an opposite tendency. We can conclude that lower BNDF/MMP-9 ratio levels may predict patients with a favorable prognosis. Our results confirmed the existence of a direct connection between the pro-plastic molecules BDNF and MMP-9.

**Abstract:**

Purpose: The development of sensitive and non-invasive biomarkers for the early detection of CRC and determination of their role in the individual stages of CRC. Methods: MMP-9 expression in serum and tissue, and BDNF expression in plasma were detected using the ELISA method. MMP-9 and BDNF in the tissue were also determined by immunohistochemical staining. Results: To assess the balance between changes in survival and tumor progression, we compared BDNF/MMP-9 ratios in tissues of living and deceased individuals. The tissue BDNF/MMP-9 ratio (evaluated immunohistochemically) decreased significantly with the progression of the disease in living patients. The BDNF/MMP-9 ratio was statistically significantly reduced in stages II and III compared to the benign group. However, in deceased individuals, the ratio showed an opposite tendency. Conclusion: The determination of the tissue BDNF/MMP9 ratio can be used as a prognostic biomarker of CRC.

## 1. Introduction

Colorectal cancer (CRC) is the most common gastrointestinal cancer and has a high mortality rate. Statistics confirm that it is one of the most prevalent cancers worldwide and is the second major cause of death related to cancer in the world [1,2,3,4,5,6]. The incidence of CRC varies from country to country. An increased incidence has been found in countries with a high Human Development Index (HDI) [7].

Since a delayed diagnosis may lead to an acceleration of mortality due to CRC, early examinations and analyses are essential. A prompt diagnosis within the initial stage and immediate treatment may prolong survival. Nowadays, the reported 5-year survival rate of early detected carcinoma is 90% and drops to 14% if the CRC is at an advanced stage [7]. Colonoscopy is still the gold-standard diagnostic tool, and chemotherapy, immunotherapy, radiotherapy, and surgery comprise the current most important strategies in CRC treatment [6,8,9,10,11,12].

The growing effort to define the correct diagnosis as soon as possible is expanded by diagnostic options that can determine specific, sensitive, and non-invasive biomarkers. Several potential clinical biomarkers, including proteins of extracellular matrix (ECM), lysyl oxidase, a disintegrin and metalloproteinases (ADAMs), thrombospondin, brain-derived neurotrophic factor (BDNF), and matrix metalloproteinases (MMPs), have been detected in tumor tissues [13]. The latter belongs to the family of zinc-dependent multidomain endopeptidases capable of degrading numerous ECM components under various physiological and pathological conditions [14,15,16]. MMPs and their endogenous inhibitors (tissue inhibitors of metalloproteinases, TIMPs) have been found to play a relevant role in colorectal cancer invasion and progression [17,18]. Recent studies have referred to significantly increased serum or tissue levels of individual MMPs in CRC. MMPs were initially thought to be involved in the formation of metastases by causing the breakdown of physical barriers by degrading ECM proteoglycans and matrix glycoproteins. However, MMPs are now known to play a role in all steps of tumor progression, affecting multiple biological functions, including modification of signaling pathways [19,20,21,22].

Cell invasion is an essential process of tumor metastasis and requires, in addition to increased expression of MMPs, another type of protease such as plasmin. Plasmin, which is a key mediator of fibrinolysis, is also significantly involved in ECM degradation. The cleavage of plasminogen to active plasmin is mediated by internal (factor XII, prekallikrein, kininogen) or external (such as tissue plasminogen activator (tPA) and urokinase plasminogen activator (uPA)) activators; uPA is localized on the cell surface through a plasminogen activator receptor (uPAR) [23,24]. The activity of uPA and tPA is regulated by plasminogen activator inhibitors (PAI-1, PAI-2). The complex interaction between plasminogen activators and inhibitors determines the extent of ECM remodeling, fibrinolysis, growth factor activation and release, tumor invasion, and metastasis. Active plasmin can subsequently cleave inactive forms of MMP (-1, -2, -3, -9, -14), which promotes further ECM degradation [25,26].

MMP-9 belongs together with MMP-2 to the subgroup of MMPs named gelatinases, as their preferred substrate is gelatin [27]. MMP-9 is synthesized as a pre-proenzyme. Then, it is secreted into the extracellular environment as a pro-enzyme by different cell types. After removing the N-terminal pro-peptide region from pro-MMP-9 via some other proteases, MMP-9 is activated [28,29]. MMP-9 is involved in many biological processes, such as proteolytic degradation of ECM, alteration of cell–cell and cell–ECM interactions, cleavage of cell surface proteins, and cleavage of proteins in the extracellular environment. MMP-9 also plays a very important role in the process of carcinogenesis when it participates in angiogenesis by regulating the expression of proangiogenic vascular endothelial growth factor [30,31,32].

BDNF is a member of the neurotrophin family of secreted growth factors with the ability to mediate through TrkB [33], resulting in the activation of several intracellular pathways affecting cell growth, proliferation, survival, angiogenesis, epithelial–mesenchymal transition (EMT), and differentiation [34,35]. TrkB has been revealed in CRC and BDNF has been reported to be expressed in over 93% of colorectal tumors [36]. BDNF is generated from a single protein precursor and the process of BDNF maturation represents the sequential cleavage of the precursor form into a mature molecule. Cleavage can take place either intra- or extra-cellularly, with MMPs (MMP-3, MMP-7, MMP-9) playing a crucial role in the latter [37]. Many authors describe the important role of MMP-9 in the activation of BDNF from pro-BDNF to its biologically active mature form and also the role of BDNF in the activation of MMP-9 gene transcription [38,39]. Although MMP2 has similar enzymatic properties to MMP9, it is not involved in BDNF maturation [40].

In our study, we tested the hypothesis that BDNF can regulate MMP-9 production in the early stages and progression of colorectal cancer.

## 2. Material

The analyzed sample consisted of 84 patients admitted to the 1st Surgical Clinic of the UNLP in Kosice, Slovakia for colon or rectal surgery. Patients were first histologically categorized according to the finding on the benign tumor group (29 patients; 34.5%) and the malignant tumor group (55 patients; 65.5%). Patients with malignant tumors were then histologically divided into stages according to the standard TNM scale. The characterization of all patients is shown in Table 1.

The morning before surgery, fasting serum and whole blood were collected from patients in a BD Vacutainer tube with separation gel for serum and K_2_EDTA for whole blood. The serum was attained after centrifugation at 3500× *g*/4 min/RT. The plasma was obtained by centrifugation of the whole blood at 2000× *g*/10 min/4 °C.

Tumor tissue (primary epithelial tumor) was removed during surgery, homogenized in extraction buffer (Invitrogen), centrifuged at 18,000× *g*/20 min/4 °C, and the supernatant was frozen at −80 °C. All subjects gave their informed consent for inclusion before they participated in the study. The study was conducted following the Declaration of Helsinki, and the protocol was approved by the Ethics Committee, 2020/EK/06042.

## 3. Methods

IBM SPSS Statistics 23 was used to create Kaplan–Meier tests and the log-rank test was used to determine differences between the survival curves.

MMP-9 expression in serum and tissue, and BDNF expression in plasma were detected by the ELISA method (Human Matrix Metalloproteinase-9 ELISA; BioVendor, Brno, Czech Republic; CS18-002S14; BDNF Human ELISA Kit; Abcam, Cambridge, UK; ab99978). The Kolmogorov–Smirnov and Shapiro–Wilk tests of normality indicated that the values were not normally distributed in individual groups. A non-parametric Kruskal–Wallis test was performed followed by post hoc analysis with Mann–Whitney U tests used to identify the specific differences between groups. A *p*-value of less than 0.05 was considered statistically significant.

Immunohistochemical staining of MMP-9 and BDNF was done on 10% formalin for 72 h, and then paraffin-embedded and cut into sections with a 4-µm thickness. The sections were incubated with a primary monoclonal antibody (MMP-9 (D6O3H XP^®^ Rabbit mAb #13667, 1:200)) and recombinant rabbit monoclonal anti-BDNF antibody (EPR1292, ab108319, 1:500) at 4 °C overnight and were stained with appropriate secondary antibodies.

The assessment of immunohistological slides followed the Breslow scale’s four classes as described previously [41]. Briefly, after scanning the entire section at medium magnification, a minimum of 500 cells were counted using high-power fields (40× objective lens). The number of MMP-9- and BDNF-positive cells was statistically evaluated, and the ratio was set.

## 4. Results

### 4.1. Kaplan–Meier Survival Analysis

In our work, we used the Kaplan–Meier univariate analysis with the log-rank test to assess the overall survival of patients, which is used to measure the number of subjects who survived or died during a certain period. The 3-year OS rate of all patients was 32.54 (95% CI, 29.82–35.26) (Figure 1A).

Furthermore, by comparing individual groups histologically stratified by stage, this analysis revealed that the third clinical stage represented a lower OS rate or a shorter median survival time compared to the benign tumor group (26.05 vs. 36.09) with *p* = 0.001 (Figure 1B), although the graph shows that, with increasing stage, the survival rate does not decrease linearly. This is possibly because the tissue samples were taken during the COVID-19 pandemic and in some patients (mainly in the first stage) it is not clear whether the cause of death was COVID-19 or colorectal cancer.

Regarding the survival of patients with different localization of the tumor, the 3-year OS rate of patients with colon cancer was 30.16 and with rectum cancer 33.1 (Figure 1C).

Kaplan–Meier survival analysis showed that the average survival time for females was significantly longer than for males (36.54 vs. 30.30) with *p* = 0.011 (Figure 1D).

### 4.2. Analyses of MMP-9 and BDNF Using ELISA

#### 4.2.1. Tissue MMP-9 in Benign/Malignant Tumor Groups

By comparing the groups of malignant and benign tumors using the ELISA method, it was found that the expression of MMP-9 is higher in the tissues of malignant tumors (Table 2). The mean of MMP-9 expression in the tissue of the benign tumor group was 176.09 ng/mL. In contrast, the mean of MMP-9 expression in the tissue of the malignant tumor group was 321.05 ng/mL (*p* > 0.05). Gender differences were noted in this baseline distribution, with higher tissue expression of MMP-9 in females. In the tissues of the benign tumor group, an average MMP-9 expression of 277.68 ng/mL was detected in females, while it was 114.27 ng/mL in males. In the malignant tumor group, the average expression of MMP-9 in the tissue of females was 355.04 ng/mL and in males 287.6 ng/mL. The differences in MMP-9 expressions in tissues between groups (benign tumor, malignant tumor) were insignificant in both sexes (*p* > 0.05).

When the malignant tumor group was divided according to the TNM scale into individual stages, it was shown that the expression of MMP-9 in tissues increased from stage I to stage II and, in stage III, the expression decreased (Figure 2A). In stage I, the mean value of MMP-9 expression in tissues was 320.39 ng/mL in females and 241.90 ng/mL in males. In stage II, it was 483.58 ng/mL in females and 347.03 ng/mL in males. Finally, in stage III, the mean value of MMP-9 expression in tissues decreased to 167.26 ng/mL in females and 235.49 ng/mL in males.

All groups of patients were also divided according to survival. The highest mean tissue expression of MMP-9 (549.48 ng/mL) was recorded in the stage II group of surviving patients and the lowest mean expression was in deceased patients (24.67 ng/mL) at this stage (Figure 2B).

The benign tumor group was divided into four subgroups according to diagnoses: patients with tubular adenoma, tubulovillous adenoma, diverticulitis, and hemorrhoids. The mean values of tissue MMP-9 expression were 4.36 ng/mL in tubular adenoma, 198.94 ng/mL in tubulovillous adenoma, 360.94 ng/mL in diverticulitis, and 233.06 ng/mL in hemorrhoids.

The highest expression of MMP-9 was recorded in the tissues of patients with a localized tumor in the sigmoid region. The mean values were 680.37 ng/mL in living patients and 359.46 ng/mL in deceased patients.

Kaplan–Meier survival analysis (Figure 3) showed that the average survival time for patients with MMP-9 < 100 ng/mL was longer than for patients with MMP-9 > 100 ng/mL (32.12 vs. 14.51; *p* > 0.05) (Figure 3).

#### 4.2.2. Serum MMP-9 in Benign/Malignant Tumor Groups

Comparing the malignant tumor group and the benign tumor group, the benign tumor group was found to have higher expression of MMP-9 in serum. The mean expression of MMP-9 in serum was 297.21 ng/mL in the benign tumor group and 117.73 ng/mL in the malignant tumor group (Table 2). This difference is statistically insignificant (*p* > *0*.05). The mean expression of MMP-9 in the serum of deceased patients in the benign tumor group was 320.07 ng/mL. In contrast, the mean expression of MMP-9 in the serum of deceased patients in the malignant tumor group was 42.33 ng/mL. However, this difference is not statistically significant (*p* > 0.05). When the malignant tumor group was divided according to the TNM scale into individual stages, expression of MMP-9 in the serum of males decreased continuously with the growing stage. The mean of MMP-9 expression in serum was 27.2 ng/mL in females and 616.14 ng/mL in males in stage I, 33.72 ng/mL in females and 148.13 ng/mL in males in stage II, 5.81 ng/mL in females and 47.29 ng/mL in males in stage III. There was no statistically significant difference within the stages (*p* > 0.05).

The expression of MMP-9 in patients with a malignant tumor compared to the expression in patients with a benign finding was lower in both sex groups. The difference in serum MMP-9 expression between groups was not significant for both sexes (*p* > 0.05).

In the benign tumor group, the mean expression of MMP-9 in serum was 196.23 ng/mL in females and 506.49 ng/mL in males. In the malignant tumor group, the mean MMP-9 expression in serum was 26.97 ng/mL in females and 161.60 ng/mL in males.

#### 4.2.3. BDNF in Benign/Malignant Tumor Groups

Comparing the malignant tumor group and the benign tumor group, the malignant tumor group was found to have higher BDNF expression in plasma. The average BDNF expression in plasma was 23.39 ng/mL in the benign tumor group and 45.86 ng/mL in the malignant tumor group. This difference is statistically insignificant (*p* > 0.05).

The mean expression of BDNF in the plasma of deceased patients in the benign tumor group was 13.83 ng/mL. In contrast, the mean expression of BDNF in the plasma of deceased patients in the malignant tumor group was 45.03 ng/mL. However, this difference is not statistically significant (*p* > 0.05). The remaining question is whether the patients died due to CRC or COVID-19 infection.

After grouping by gender, BDNF plasma expressions were found to be higher in the malignant tumor group in both sexes. The difference in expression of BDNF in plasma between groups (benign tumor, malignant tumor) was insignificant in both sexes (*p* > 0.05). The expression of BDNF in the plasma of females was 24.34 ng/mL in the group with benign tumors and 49.66 ng/mL in the group with malignant tumors. In males in the group with a benign tumor, the average expression of BDNF in plasma was 11.27 ng/mL and, in the group with a malignant tumor, 43.32 ng/mL. The mean BDNF expression in the benign tumor group and in the individual stages of CRC are shown in Figure 4A.

The mean plasma BDNF expression in the benign tumor group was 23.39 ng/mL for all patients, 26.26 ng/mL for living patients, and 13.83 ng/mL for patients who died. The mean expression in all stage I CRC patients was 59.18 ng/mL, in living patients 57.98 ng/mL, and in deceased patients 37.16 ng/mL. The mean value of BDNF expression in plasma of stage II patients was 40.31 ng/mL in all patients, 42.6 ng/mL in living patients, and 35.74 ng/mL in deceased patients. Finally, the mean expression of BDNF in plasma of stage III patients was 41.05 ng/mL in all patients, 5.34 ng/mL in living patients, and 51.26 ng/mL in deceased patients (Figure 4B). The statistical significance between the individual groups using the Kruskal–Wallis test was not statistically significant (*p* > 0.05).

The benign tumor group was divided into four subgroups according to diagnoses: patients with tubular adenoma, tubulovillous adenoma, diverticulitis, and hemorrhoids. The average value of BDNF expression in the plasma of tubular adenoma patients was 8.69 ng/mL, in tubulovillous adenoma patients 25.74 ng/mL, in diverticulitis patients 65.74 ng/mL, and in hemorrhoids patients 13.62 ng/mL. The statistical difference using the Kruskal–Wallis test was insignificant (*p* > 0.05).

The mean value of BDNF expression in the plasma of living patients with rectal tumors was 37.9 ng/mL and 40.4 ng/mL of living patients with sigmoid tumors. The average value of BDNF expression in deceased patients with rectal tumors was 9.59 ng/mL, with sigmoid tumors 56.2 ng/mL, and with rectosigmoid tumors 51.51 ng/mL.

Kaplan–Meier survival analysis (Figure 5) showed that the average survival time for patients with BDNF < 30 ng/mL was longer than for patients with BDNF > 30 ng/mL (32.45 vs. 12.89; *p* > 0.05) (Figure 5).

### 4.3. Analyses of MMP-9 and BDNF Using Immunohistochemical Staining

#### 4.3.1. Determination of Tissue MMP-9 Expression by IHC

The expression of MMP-9 in individual tissue samples was confirmed by immunohistochemistry (Figure 6). The results of IHC differed from the results recorded by the ELISA method, where the highest expression of MMP-9 was in living patients in stage II of CRC (Table 3).

Immunohistochemical analysis revealed greater expression of MMP-9 in malignant tissue compared to tissue with benign findings. In all stages, there were significant differences in MMP-9 expression between surviving patients and those who were deceased (Table 3). In patients who died, the expression of MMP-9 was much higher but surprisingly decreased with increasing stage of CRC.

Like MMP-9, the BDNF expression in tissue was confirmed by immunohistochemical staining (Figure 7). Immunohistochemical analysis revealed that, in stage I and stage II, there were higher mean values of BDNF expression in living patients compared to deceased patients, but in stage III it was the opposite and there was higher expression of BDNF in patients who were deceased. In a general comparison of benign and malignant patients, BDNF expressions were lower in the group with benign findings (Table 3). To evaluate the statistical significance of the IHC method, we used Sidak’s multiple comparison test.

#### 4.3.2. BDNF/MMP-9 Ratios

To assess the balance between changes in survival and tumor progression, we compared BDNF/MMP-9 ratios both in tissues, and in the blood of living and deceased individuals to reveal the potential diagnostic value of this parameter. As shown in Figure 8A, the tissue BDNF/MMP-9 ratio (evaluated immunohistochemically) decreased significantly with the progression of the disease in living patients. The BDNF/MMP-9 ratio was statistically significantly reduced in stages II and III compared to the benign group (*p* < 0.05 and *p* < 0.001, respectively) in patient tissues using Sidak’s multiple comparison tests. Interestingly, stage III tumor patients had a significantly decreased BDNF/MMP-9 ratio related to stage I CRC (*p* < 0.05). However, in deceased individuals, the ratio showed an opposite tendency. With the increasing stage of CRC, the ratio of BDNF/MMP-9 increased significantly. We observed serious differences in the BDNF/MMP-9 ratio of stage III against stage I in the group of patients with benign findings (*p* < 0.01) and stage I (*p* < 0.05).

BDNF/MMP-9 ratios (Figure 8B) determined in blood did not correlate with the ratio in tissue. In the group of living patients, the ratio decreased in stage I patients compared to patients with benign findings, but the ratio increased again in stage II patients. In the group of deceased patients, the trend was similar to living patients, with the highest ratio in stage III patients.

## 5. Discussion

Surgery dominates in the treatment of colorectal cancer. The surgical procedure itself is the basis of the curative treatment of colorectal cancer. However, early detection of less advanced colorectal cancer is a crucial prerequisite for achieving a better patient outcome. Patient survival depends on the stage of the disease at the time of diagnosis or at the time of surgery, but, currently, over 30% of patients are at an advanced stage at the time of diagnosis, where surgery becomes only an optional part of complex palliative treatment [42,43].

Since a significant portion of patients relapse after surgery due to the spread of the disease (formation of distant metastases), the basic treatment method for patients in an advanced stage of the disease (recurring and advanced at the time of diagnosis) remains chemotherapy, radiotherapy, or immunotherapy [44,45].

Opinions on the benefit of adjuvant chemotherapy in clinical stage II vary and an individual approach is preferred. Approximately 20% of patients in stage II of the disease experience disease relapse within five years after curative surgery. Knowing the subcellular level of the essence of the tumor cell has a fundamental impact on the choice of therapeutic procedures [46,47].

The male-to-female ratio among patients with CRC in the current study was 1.63, which is slightly higher compared to the study by Fang et al. [3] where this ratio was 1.5. If we compare the male-to-female ratio in individual stages, 4× more males were diagnosed with cancer in stage III and 3× more males were diagnosed with cancer in stage II.

A study by Fang et al. in China showed that 53.5% of patients were diagnosed with colon cancer, whereas our findings showed that the average proportion of patients with colon cancer was even 67%.

The proportion of patients with advanced cancer (stage III) in our study was 35%, which was approximately the same as in the studies by Fang et al. or Shi et al. [3,48].

Colorectal EMT is controlled by a complex network of signaling pathways involving various regulators. Matrix metalloproteinases (MMPs), especially MMP-9, are considered central signaling hubs of EMT and major promoters of CRC progression, invasion, and metastasis. MMP-9 may serve as a potential therapeutic target in stopping the spread of colorectal tumors and the formation of metastases [17,49,50,51].

Our results from tissue samples confirm that the most important role is played by MMP-9 in stage II, when their expression in living patients is the highest, but in contrast to deceased patients where is the lowest. At this stage, cancer has penetrated through the wall of the colon or rectum, which supports the hypothesis that MMP-9 levels are associated with cancer progression [35]. Consequently, timely surgical intervention becomes crucial at this particular stage of cancerogenesis. Adequate surgical resection of the primary tumor offers the best chance at remission. In patient management, it is not reasonable to rely on subsequent radiation or chemotherapy if the resection is not successful. Thus, overexpression of MMP-9 may serve as an indicator of both poor survival rates and high recurrence rates. However, a limitation in our experiment was that MMP-9 expressions were not determined by patients both before treatment and as a monitoring of treatment success. Despite this, Kaplan–Meier survival analysis demonstrated that patients with MMP-9 levels below 100 ng/mL had a longer average survival time compared to those with MMP-9 levels above 100 ng/mL.

Regarding the determination of MMP-9 expression in blood serum, our results in patients with benign and malignant tumors are in contradiction with many authors who previously reported significantly higher corresponding values in a malignant tumor group [28,52,53].

This discrepancy could be attributed to the fact that our patient groups solely consisted of individuals with benign conditions, such as hemorrhoids, diverticulitis, and adenomas, which are essentially acute inflammatory diseases. In these conditions, matrix metalloproteinases play a significant role and, during the acute phase, their expression in the bloodstream is elevated.

Nevertheless, our findings align with the results presented by Otero-Estévez et al. [21], who conducted a cohort study comparing malign and benign forms of colorectal carcinoma. Their conclusions describe a decrease in serum MMP-9 concentrations in the malignant group compared to the benign tumor group. They also emphasize the influence of gender and age on MMP-9 values, suggesting that these parameters should be monitored to ensure accurate analysis. According to their study, even though serum MMP-9 concentrations have diagnostic value, the diagnostic accuracy is lower than that of a FIT (fecal immunochemical test).

In tumor development and progression, growth factors called neurotrophins (NT) and their tyrosine kinase receptors (tropomyosin receptor kinase (Trk) are commonly implicated. Among them, brain-derived neurotrophic factor (BDNF) stands out as a critical and extensively studied neurotrophin. The expression of BDNF regulates cellular proliferation and survival, and is also associated with cell invasion through the secretion of matrix metalloproteinases, particularly MMP-9 [54].

Our findings affirm that plasma BDNF expressions are elevated in the malignant tumor group in comparison with the benign tumor group. These results align with previous studies conducted by several authors, who have attributed significant roles to this protein in the development and progression of various types of cancer, including CRC [55,56,57,58].

However, we observed a notable disparity between our findings and those reported by Brierley et al. [59]. Their study revealed a significantly lower median value of serum BDNF expression in CRC patient samples. However, it is important to note that the authors compared malignant patients with healthy controls, which introduces a distinct context for their analysis.

Although no statistically significant difference was observed in plasma BDNF concentrations between deceased patients in the benign and malignant tumor groups, these findings hold promise for further investigation. They align with the study conducted by Junior et al. [60], which reported that increased BDNF expression in CRC cases correlated with enhanced invasiveness, tumor cell migration, and the occurrence of local and liver metastases. Thus, BDNF expression is associated with a poor patient prognosis, which essentially correlates with our results in patients who died. In these patients, BDNF levels increased with the progression of the disease. In contrast, our results for surviving patients highlight the crucial role of BDNF in stage I, where its association with proliferation and invasion of colorectal cell carcinoma becomes most significant. The Kaplan–Meier test results of our study showed that the average survival time for patients with BDNF < 30 ng/mL was longer than for patients with BDNF > 30 ng/mL.

Based on these observations, we established the BDNF/MMP-9 ratio, which yielded intriguing results when assessed through immunohistological measurements and expression measurements using the ELISA method. The IHC ratios of BDNF/MMP-9 demonstrated statistically significant findings. In surviving patients, there was a decreasing trend of the ratio with advancing stages, while, in the group of patients who died, the BDNF/MMP-9 ratio increased as the stage progressed. The ratio values in the deceased patient group were lower than those in the group of surviving patients.

Thus, BDNF/MMP-9 ratio could be a suitable marker for distinguishing stages as well as predicting survival in patients with colorectal cancer. We can conclude that lower BNDF/MMP-9 ratio levels may predict patients with a favorable prognosis. While multiple factors have been described to regulate MMP-9 expression in different types of disease, the molecular mechanism that controls MMP-9 transcription in colorectal cancer remains poorly understood. Our results confirmed the existence of a direct connection between the proplastic molecules BDNF and MMP-9.

Despite these promising results, we acknowledge certain limitations of this study. To achieve more robust statistical significance, it will be necessary to expand the size of individual patient groups in future investigations. Additionally, long-term health monitoring of the patients involved in this study is imperative to ensure the reliability and validity of our findings.

## 6. Conclusions

Regardless of the progress in CRC treatment outcomes, stage II CRC remains a challenge in terms of curative strategies; therefore, there is still a need to find suitable biomarkers that could identify high-risk patients with poor prognoses. Our patients with stage II CRC have a 5-year survival of 79%, which drops to exactly 35% if the tumor has already been classified as stage III.

The ratio of tissue BDNF/MMP-9 is significantly higher in living patients and decreases with tumor progression. Nevertheless, as a result of differences between patients in individual groups, a personalized approach to patients is necessary. However, the determination of tissue BDNF/MMP-9 ratio as prognostic biomarkers of CRC can be used.

## Figures and Tables

**Figure 1 biomedicines-11-01839-f001:**
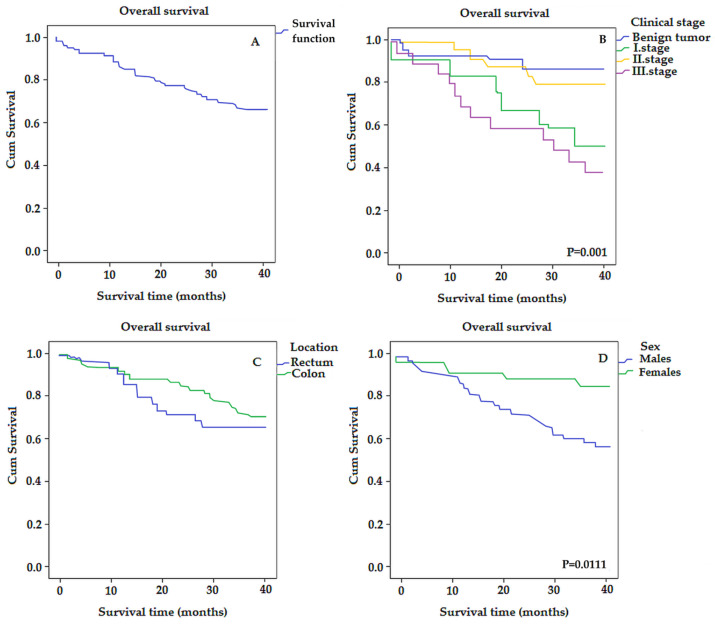
The Kaplan–Meier survival curves for colorectal-cancer-specific mortality in all patients (**A**), and according to individual stages (**B**), location (**C**), and gender (**D**).

**Figure 2 biomedicines-11-01839-f002:**
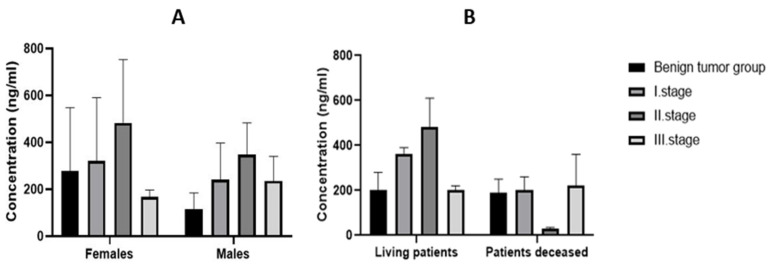
Tissue expression of MMP-9 in the individual stages of CRC (**A**) according to gender and (**B**) according to survival.

**Figure 3 biomedicines-11-01839-f003:**
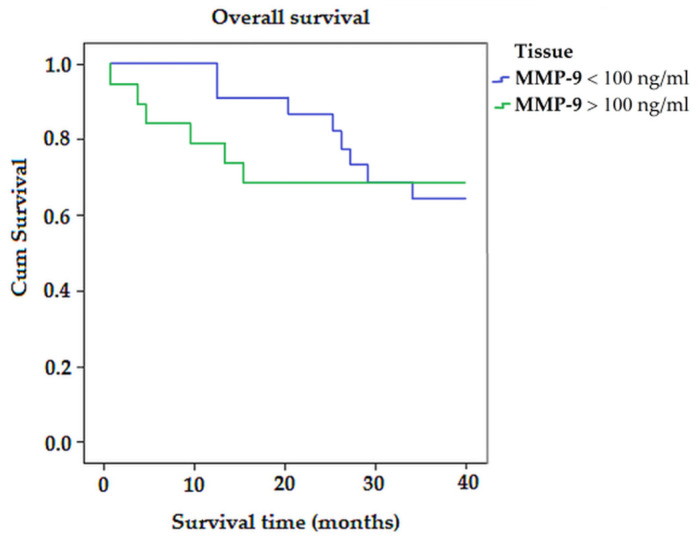
The Kaplan–Meier survival curves for colorectal-cancer-specific mortality in all patients, according to the expression of MMP-9.

**Figure 4 biomedicines-11-01839-f004:**
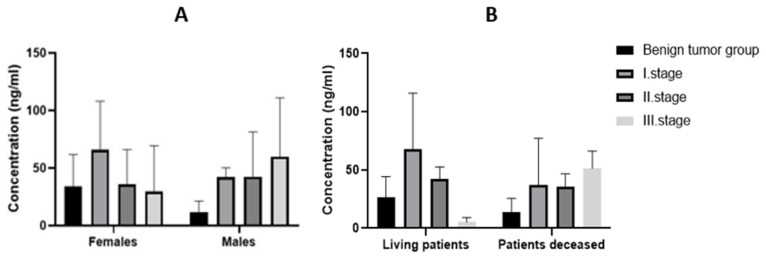
Expression of BDNF in plasma of individual stages of CRC (**A**) according to gender and (**B**) according to survival.

**Figure 5 biomedicines-11-01839-f005:**
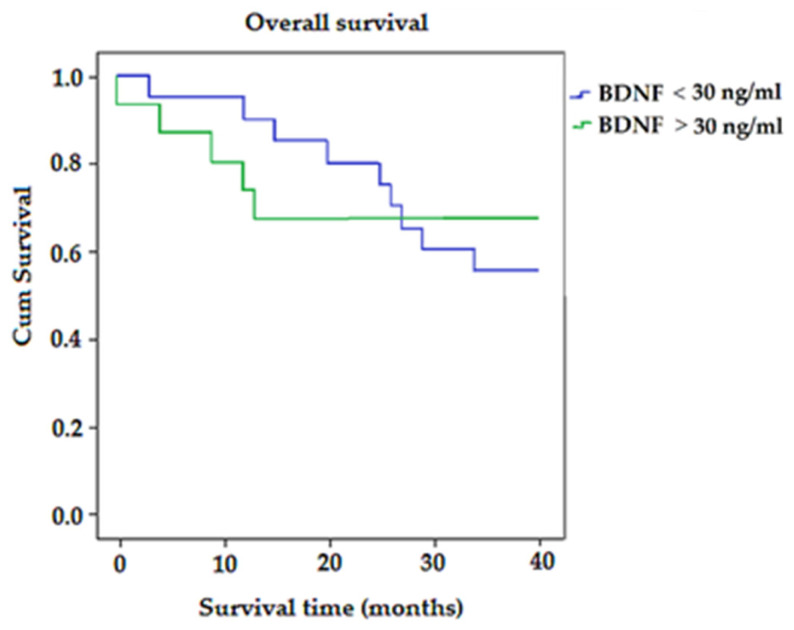
The Kaplan–Meier survival curves for colorectal-cancer-specific mortality in all patients, according to expression of BDNF.

**Figure 6 biomedicines-11-01839-f006:**
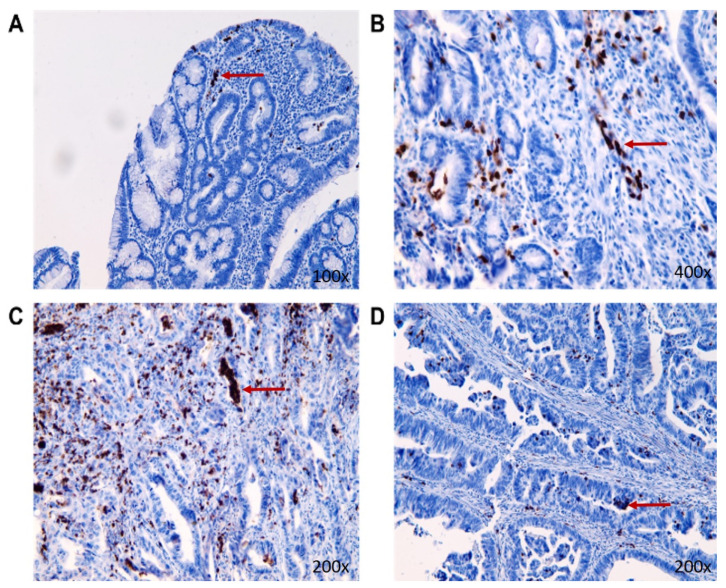
The representative pictures of MMP-9 nuclear presence (the red arrow) in the tumorous tissue of individual stages in CRC living patients ((**A**): benign tumor group; (**B**): stage I of CRC; (**C**): stage II of CRC; (**D**): stage III of CRC).

**Figure 7 biomedicines-11-01839-f007:**
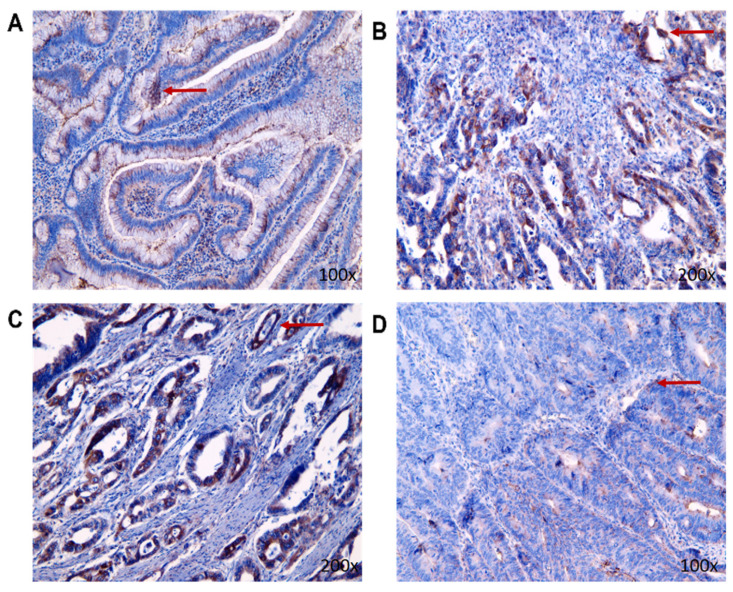
The representative pictures of BDNF nuclear presence (the red arrow) in the tumorous tissue of individual stages in CRC living patients ((**A**): benign tumor group; (**B**): stage I of CRC; (**C**): stage II of CRC; (**D**): stage III of CRC).

**Figure 8 biomedicines-11-01839-f008:**
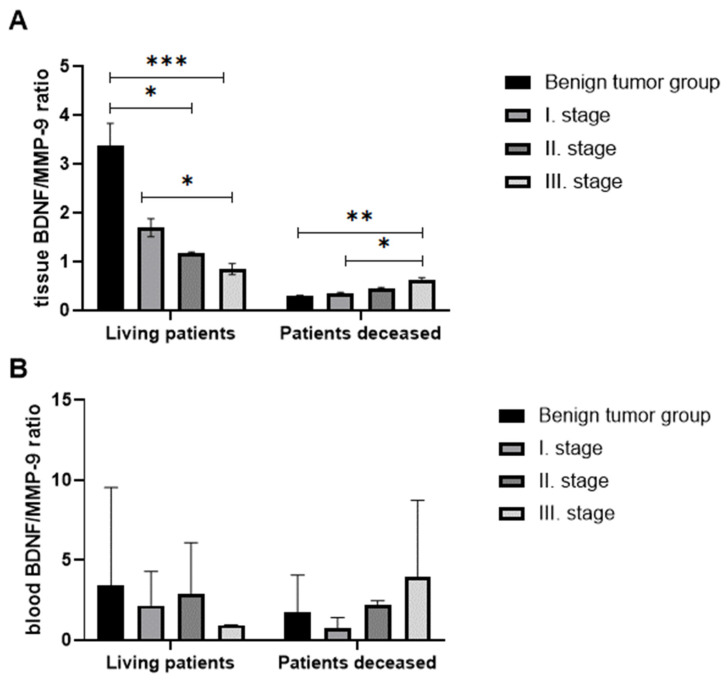
The ratio of tissue BDNF/tissue MMP-9 (**A**) and the ratio of plasma BDNF/serum MMP-9 (**B**) in living and deceased patients. (*p* < 0.05 *, *p* < 0.01 **, *p* < 0.005 ***).

**Table 1 biomedicines-11-01839-t001:** Classification of patients into groups according to individual parameters.

Number of Patients				
	Benign Tumor Group	Malignant Tumor Group		
		Stage I	Stage II	Stage III
Sex				
Males Females	12	7	18	15
	17	5	6	4
Tumor localization				
Colon	20	7	16	14
Rectum	9	5	8	5
Findings				
Hemorrhoids	4	-	-	-
Diverticulitis	5	-	-	-
Adenomas	20	-	-	-
Adenocarcinomas	-	12	24	19
Survival				
Living patients	26	6	19	7
Deceased patients	3	6	5	12

**Table 2 biomedicines-11-01839-t002:** MMP-9 levels in tissues and serum, and BFNF levels in plasma determined by ELISA method.

	Tissue MMP-9 Levels (ng/mL)	Serum MMP-9 Levels (ng/mL)	Plasma BDNF Levels (ng/mL)
Benign tumor group			
Females Males *p*	277.68 (4.39–886.19) 114.27 (3.91–586.69) >0.05	46.08 (3.83–197.64) 506.48 (6.80–1046.05) >0.05	24.34 (4.74–55.74) 11.27 (5.12–29.74) >0.05
Malignant tumor group			
Females Males *p*	355.04 (4.94–1038.41) 287.6 (4.04–1155.27) >0.05	20.63 (3.85–82.55) 283.33 (4.38–1173.02) >0.05	49.66 (0.89–120.68) 43.32 (1.49–142.11) >0.05
Benign tumor group (both sexes)	176.09 (4.35–886.19)	297.21 (3.83–1046.05)	23.39 (4.74–65.74)
Stage I			
Living patients Patients deceased *p*	279.16 (11.84–869.28) 200.21 (60.38–239.041) >0.05	320.56 (31.70–1173.02) 29.41 (3.85–50.73) >0.05	67.98 (37.92–120.68) 37.16 (3.89–70.29) >0.05
Stage II			
Living patients Patients deceased *p*	549.48 (4.06–868.17) 24.67 (19.18–33.02) 0.04 *	94.86 (5.28–364.07) no data >0.05	42.59 (11.24–67.74) 35.74 (4.25–96.54) >0.05
Stage III			
Living patients Patients deceased *p*	233.14 (46.37–272.14) 254.61 (4.04–784.15) >0.05	9.70 (1 patient) 46.64 (4.38–245.86) >0.05	5.34 (2.58–9.18) 51.26 (3.33–142.12) 0.008 **

*p* < 0.05 *, *p* < 0.01 **.

**Table 3 biomedicines-11-01839-t003:** Data obtained by immunohistological method quantification.

	IHC BDNF	IHC MMP-9
Benign tumor group	101 (98–103)	97 (89–104)
Stage I		
Living patients Patients deceased *p*	188 (155–208) 106 (102–116) 0.004 **	111 (99–115) 314 (282–335) 0.004 **
Stage II		
Living patients Patients deceased *p*	143 (128–151) 121 (115–130) 0.0051 **	125 (110–130) 275 (262–285) 0.002 **
Stage III		
Living patients Patients deceased *p*	109 (101–118) 151 (145–160) 0.002 **	132 (117–145) 242 (228–251) 0.004 **

*p* < 0.01 **.

## Data Availability

The data are not available due to the protection of personal data.

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
