# Peer review of "Stage-Dependent Levels of Brain-Derived Neurotrophic Factor and Matrix Metalloproteinase 9 in the Prognosis of Colorectal Cancer"

_biomedicines, 2023, doi:10.3390/biomedicines11071839_

Round 1

Reviewer 1 Report (Previous Reviewer 1)

Manuscript entitled "Stage-dependent levels of brain-derived neurotrophic factor and matrix metalloproteinase 9 in the prognosis of colorectal cancer"

The authors claimed that the majority of MMP-9 in the benign group was in an inactive form, and in the group of patients with adenocarcinomas the concentration of active forms increases as the stage increases. As for BDNF values, they increased in the group of patients with CRC, but only in patients who died. Fonally, they claimed that the determination of the BDNF/MMP9 ratio can be used as a prognostic biomarker of CRC.

Major issues:

1. I siggest the authors should measure BDNF/MMP9 ratio by means of IHC in a well collected cohort to see the prognostic significance in a tiome-to-event manner.

2. For the evaluation of BDNF/MMP9 ratio, double stain should be used.

acceptable

Author Response

  1. I suggest the authors should measure BDNF/MMP9 ratio by means of IHC in a well collected cohort to see the prognostic significance in a time-to-event manner. 

- Thank you very much for your opinion. As you mentioned, it may be of higher interest to compare the BDNF/MMP9 ratio in IHC sections. So we did the BDNF/MMP9 ratios extra for blood and for tissue (by mean of IHC) and added the results to the manuscript (see labeled in red). 

  1. For the evaluation of BDNF/MMP9 ratio, double stain should be used. 

- We see the point that you mean. But bad luck, both primary antibodies (as you can see in Methods section) are of the rabbit origin, so we cannot provide the double staining. In our country, it is impossible to get primary antibody in few days, it takes usually months long. We asked also other laboratories, but they don´t have other ones. So we have to say sorry that we cannot provide the double stained samples.  

Reviewer 2 Report (New Reviewer)

The authors examined the clinical status of BDNF and MMP9 in cases of colorectal cancer.

Although it is worthwhile to examine the candidate indexes between groups (stage, presence or absence of metastases, etc.) according to the clinical condition...prognostic factors, etc., would still need to be examined by Kaplan-Meier. As for the distinction between dead and alive cases, well, it is a good idea to try, but it is too superficial to consider everything on that basis alone.

In addition, what will be the biological significance of the ratios observed for both factors in this study?

These should be improved.

Author Response

First of all, we would like to thank reviewer for providing us with positive feedback and constructive comments regarding our article, entitled Stage-dependent levels of brain-derived neurotrophic factor and matrix metalloproteinase 9 in the prognosis of colorectal cancer.

We have addressed all the comments made by the reviewer, and we honestly feel that the quality of the article has improved as a consequence.

We provide below a detailed reply to all the reviewer comments.

Comments and Suggestions for Authors

The authors examined the clinical status of BDNF and MMP9 in cases of colorectal cancer.

Although it is worthwhile to examine the candidate indexes between groups (stage, presence or absence of metastases, etc.) according to the clinical condition...prognostic factors, etc., would still need to be examined by Kaplan-Meier. As for the distinction between dead and alive cases, well, it is a good idea to try, but it is too superficial to consider everything on that basis alone.

Answer: Thanks for the great guidance on using the Kaplan-Meier test for patient survival. We sincerely observe that the article has improved significantly

In addition, what will be the biological significance of the ratios observed for both factors in this study?

Answer: We evaluated the ratios in a slightly different way and prepared the ratios of both parameters based on individual techniques. We discussed the biological significance of the ratios better in the discussion section

Reviewer 3 Report (New Reviewer)

Manuscript No Biomedicines-2433595

Stage-dependent levels of brain-derived neurotrophic factor and matrix metalloproteinase 9 in the prognosis of colorectal cancer” for Biomedicines

Comments:

1. Title. Please remove the dot after the title.

2. Affiliations. Affiliation No. 3. The name of the Department should be capitalized.

3. In the introduction, please also mention the uPA/uPAR system as an important parameter related not only to matrix remodeling but also to the activation of, for example, MMPs.

4. Please explain in the introduction why MMP-9 was analyzed and not MMP-2 or both in parallel.

Author Response

First of all, we would like to thank reviewer for providing us with positive feedback and constructive comments regarding our article, entitled Stage-dependent levels of brain-derived neurotrophic factor and matrix metalloproteinase 9 in the prognosis of colorectal cancer.

We have addressed all the comments made by the reviewer, and we honestly feel that the quality of the article has improved as a consequence.

We provide below a detailed reply to all the reviewer comments.

  1. Please remove the dot after the title.

Answer: The dot has been removed

  1. Affiliation No. 3. The name of the Department should be capitalized.

Answer: The department name has been corrected in capital letters.

  1. In the introduction, please also mention the uPA/uPAR system as an important parameter related not only to matrix remodeling but also to the activation of, for example, MMPs.

Answer: The uPA/uPAR system was explained in the introduction, line 67-78

  1. Please explain in the introduction why MMP-9 was analyzed and not MMP-2 or both in parallel.

Answer: MMP-2 does not play any role in the maturation of BDNF, which was mentioned in the introduction, line 98

Reviewer 4 Report (New Reviewer)

In this paper, Večurkovská and co-workers analyze the correlation between the levels of BDNF and MMP-9 in the prognosis of CRC. There are some interesting results here, both the objective of the paper and collateral data. An example of the later is the fact that female subjects express higher levels of MMP9 compared to males. The authors attempt to create a ratio of BDNF and MMP9 to determine the prognosis of CRC patients.

Overall, this paper reads like a missed opportunity. The authors have access to an important pool of stratified CRC patients, but the results are not convincing. Conclusions are not decisive. For example, it seems that active-MMP-9 / pro-MMP-9 ratio is somewhat similar between live and deceased patients, but the trend is similar. In this regard, in Fig. 12, is the BDNF/MMP-9 ratio calculated according to total MMP-9? Or is it active? Why do the author think the ratio declines in stage III?

There's not enough information regarding the histology of the tumors. Are all epithelial? Is any of them metastatic?

Some of the data (e.g. Fig. 4 Fig. 8 and 9) are illustrations, that is, representative data; however, these data need to be properly quantified. In this regard, data presentation is too sparse and readers would benefit from tighter presentation (12 figures are too much for this type of paper).  

In general, the paper is poorly written, and the organization of the data is confusing. The authors are advised to use data-oriented titles, instead of molecule-oriented (for example, Active/pro-forms of MMP-9 should be something like “MMP-9 activity correlates with patient staging”). Also, a native English speaker needs to look at the paper.

The paper requires additional revision by a native speaker.

Author Response

Začiatok formulára

In this paper, Večurkovská and co-workers analyze the correlation between the levels of BDNF and MMP-9 in the prognosis of CRC. There are some interesting results here, both the objective of the paper and collateral data. An example of the later is the fact that female subjects express higher levels of MMP9 compared to males. The authors attempt to create a ratio of BDNF and MMP9 to determine the prognosis of CRC patients.

Overall, this paper reads like a missed opportunity. The authors have access to an important pool of stratified CRC patients, but the results are not convincing. Conclusions are not decisive. For example, it seems that active-MMP-9 / pro-MMP-9 ratio is somewhat similar between live and deceased patients, but the trend is similar. In this regard, in Fig. 12, is the BDNF/MMP-9 ratio calculated according to total MMP-9? Or is it active? Why do the author think the ratio declines in stage III?

Answer:
We evaluated the ratios in a slightly different way and prepared the ratios of both parameters based on individual techniques. We discussed the biological significance of the ratios better in the discussion section

There's not enough information regarding the histology of the tumors. Are all epithelial? Is any of them metastatic?

Answer: We added more information about the histology of tumors in the materials section

Some of the data (e.g. Fig. 4 Fig. 8 and 9) are illustrations, that is, representative data; however, these data need to be properly quantified. In this regard, data presentation is too sparse and readers would benefit from tighter presentation (12 figures are too much for this type of paper).  

In general, the paper is poorly written, and the organization of the data is confusing. The authors are advised to use data-oriented titles, instead of molecule-oriented (for example, Active/pro-forms of MMP-9 should be something like “MMP-9 activity correlates with patient staging”). Also, a native English speaker needs to look at the paper.

Answer: We have significantly changed the article, removed many images, added new graphs - Kaplan-Meier survival, put extensive results into tables, and we think that thanks to your comments and recommendations, the article has become clearer

Round 2

Reviewer 1 Report (Previous Reviewer 1)

Apologize but there is no much improvement.

Reviewer 2 Report (New Reviewer)

Authors changed their manuscript according to the reviewers' comments, adequately. 

Reviewer 4 Report (New Reviewer)

The authors have addressed my concerns and questions appropriately.

This manuscript is a resubmission of an earlier submission. The following is a list of the peer review reports and author responses from that submission.

Round 1

Reviewer 1 Report

Manuscript entitled "Stage-dependent levels of brain-derived neurotrophic factor and matrix metalloproteinase 9 in the prognosis of colorectal cancer."

1. This work is relatively preliminary. The author should evaluate the biomarkers with various important clinicopathologic factors such as age, gender, differentiation, intravascular/perineurial invasion, as well as RAS mutation/dMMR status.

2. The IHC should be performed to confirm the expression of biomarkers in tumor tissue instead of other organs.

3. The prognostic significances of biomarkers should be performed in univariate and multivariate manners.

Author Response

We thank the reviewer for the review of our manuscript as well as for insightful critiques that would help us improve the manuscript. However, we are afraid we don't quite understand: your questions are intended as recommendations for further work or as modifications without which the article cannot be published? 

  1. This work is relatively preliminary. The author should evaluate the biomarkers with various important clinicopathologic factors such as age, gender, differentiation, intravascular/perineurial invasion, as well as RAS mutation/dMMR status. 

Answer: We agree that this study is preliminary, we are continuing in the differentiation of patients concerning the presence/absence of individual mutations and MMR status, as well as determining the expression of biomarkers by Western blotting or IHC methods. These further examinations will be published in another article that will follow this one. Regarding the division of patients based on age and gender, as well as invasiveness, everything is described in the manuscript. 

2. The IHC should be performed to confirm the expression of biomarkers in tumor tissue instead of other organs. 

Answer: Individual biomarkers were determined in tissues previously removed during surgery and following histopathologically determined tumor grade by TNM classification 

3. The prognostic significances of biomarkers should be performed in univariate and multivariate manners. 

Answer: We used the Friedman, a non-parametric test alternative to ANOVA, and Kruskal-Wallis test for evaluating the results. Do you recommend any other statistical tests? 

Reviewer 2 Report

In this manuscript, Ivana Vecurkovska et al showed that stage-dependent levels of brain-derived neurotrophic factor and MMP9 in the prognosis of colorectal cancer.  These findings are potentially interesting. The manuscript could be further strengthened with a few additional experiments denoted below. 

1. The authors need to explain more about MMPs in CRC patients.
2. There are many places that incorrectly or inaccurately write down the manuscript such as figure legends. Authors need to pay close attention to proper labeling of this manuscript.

3. Most of the labeling is omitted in Figure, making it difficult to know the unit.

4. In figure 5, please show the Tubular adenoma data in the figure.

5. The data is too scattered. I think it would be good to integrate some figures into one.

6. In figure 10 and 11, authors have to show the protein marker otherwise we don't know the exact size

Author Response

First of all, we would like to thank the editor and reviewers for providing us with positive feedback and constructive comments regarding our article, entitled Stage-dependent levels of brain-derived neurotrophic factor and matrix metalloproteinase 9 in the prognosis of colorectal cancer.

We have addressed all the comments made by the reviewers, and we honestly feel that the quality of the article has improved as a consequence.

We provide below a detailed reply to all the reviewers’ comments.

  1. The authors need to explain more about MMPs in CRC patients.

Response 1: We added a few sentences about MMPs and specifically about MMP-9 to the introduction.

  1. There are many places that incorrectly or inaccurately write down the manuscript such as figure legends. Authors need to pay close attention to proper labeling of this manuscript.

Response 2: Thank you for this valuable comment. We carefully checked (and corrected where needed) the labels in the text and legends for each figure.

  1. Most of the labeling is omitted in Figure, making it difficult to know the unit.

Response 3: We have added all labels and units to the images

  1. In figure 5, please show the Tubular adenoma data in the figure.

Response 4: We threw out this image because due to the small numbers we could not make a graph where these values would be visible

  1. The data is too scattered. I think it would be good to integrate some figures into one.

Response 5: You are right, the images seem too stretched. Since there are diametrically different values in the figures, some data would not be visible after merging some figures. Nevertheless, we merged fig. 3 and 4 into one - fig. 3, we threw out figure 5. We combined fig 7 and 8 into one – fig. 5

  1. In figure 10 and 11, authors have to show the protein marker otherwise we don't know the exact size

Response 6: We cut out only bands with pro-MMP-9 (92 kDa) and active MMP-9 (86 Da) from the individual zymograms. If we insert a protein marker, it will be the same as when we added the arrows with the label and size of the bands. If the reviewer still thinks that a protein marker is desirable, of course, we will insert it.

Round 2

Reviewer 1 Report

The authors didn't make modification based on review opinions. 

Reviewer 2 Report

The authors have completed all the answers to the questions I raised.

Thanks.